# Does digital transformation enhance the core competitiveness?—Quasi-natural experimental evidence from Chinese traditional manufacturing

**LiuYang Zhang**[1]\*, **PingQian Qiu**[1], **Peng Cao**[2]

**1** Business School, Hunan Normal University, Changsha, Hunan, PR China, **2** Business School, Henan University of Technology, Zhengzhou, Henan, PR China

\* leontbabm@163.com

**Data Availability Statement:** All data are available from the CSMAR database (http://cndata1.csmar. com/).

## Abstract

In the era of the digital economy, building an internationally competitive manufacturing industry with intelligent manufacturing as its main focus is the only way to promote the transformation of a country into a manufacturing power and achieve high-quality economic development. To explore whether Digital transformation can improve the core competitiveness of traditional manufacturing enterprises, and what factors affect this process, this study establishes the core competitiveness system of enterprises through the principal component analysis(PCA) method and discusses the above issues through the construction of a double difference model. The results of this research from China's traditional manufacturing industry are as follows. (i) The digital transformation of enterprises has significantly improved their core competitiveness and has a certain time lag effect. (ii) In the process of enterprise digital transformation, enterprise management capabilities, environmental uncertainty, and enterprise operational efficiency will positively enhance the results of enterprise digital transformation. (iii) The enhancement of core competitiveness caused by digital transformation is more significant for the market leaders and laggards. (iv) Compared with non-state-owned enterprises, the Digital Transformationn of state-owned enterprises has a more obvious effect on promoting their core competitiveness. (v) In comparison with enterprises with low government subsidies, the Digital Transformation of enterprises with high government subsidies plays a more significant role in promoting their core competitiveness. In addition, this study proposes policy guidance and practical guidance for digital transformation to accelerate the promotion of core competitiveness of traditional manufacturing industry.

## 1. Introduction

Given the gradual maturity and application of "big data, intelligent robots, cloud" and other technologies, the data processing ability and analysis have been upgraded from KB to PB. The resulting digital transformation is gradually leading the transformation of modern enterprises.

**Funding:** The authors received no specific funding for this work.

**Competing interests:** The authors have declared that no competing interests exist.

Especially for manufacturing enterprises, it is particularly important to complete digital transformations. As the main body of the national economy, China's manufacturing industry has contributed to nearly one-third of the economy, ranking first in the world for 12 years. However, compared with high-end manufacturing in developed countries, China's manufacturing industry still faces the problem of being "big but not strong". The lack of independent innovation has led to China's manufacturing industry as a whole being at the end of the world's value chain. The Sino-US trade frictions and the global COVID-19 pandemic in 2020 also forced Chinese manufacturing enterprises to face substantial competitive pressure, and the previously established supply chain system suffered a heavy blow. In the international context of increasingly fierce global technological and industrial competition, it is of great significance for China to transform itself into a manufacturing power by seizing the industrial transformation opportunity using digital technology as the core and making efforts towards the middle and high end of the "smile curve". Therefore, China proposes using digital transformation and digital manufacturing as the keys to achieving the high-quality development of the domestic manufacturing industry and issued the "Fourteenth Five-Year Plan" for the Development of Intelligent Manufacturing, which puts forward the need to cultivate new intelligent manufacturing formats and models at multiple levels and explore new paths for the transformation and upgrading of manufacturing enterprises. Especially in the context of the global spread of COVID-19 in 2020, digital technology plays an important role in improving the core competitiveness of traditional manufacturing.

However, digital transformation is a gradual process and is not achieved overnight. Within an enterprise, digital transformation technologies complement each other and gradually complete the process of "1+1>2". The degree of digitalization of the enterprise is also gradually increasing; however, is its core competitiveness also improving at the same time? Currently, the literature on the relationship between the two is lacking. Most existing studies have conducted research from the perspective of the impact of digital transformation on enterprise performance; however, the conclusions reached by scholars are quite different, mainly in the following four aspects: first, positive promotion [1–11]; second, reverse promotion [12, 13]; third, an inverted U-shaped impact [14–17]; and fourth, no impact [16, 17].

In summary, most existing studies start from enterprise performance and use OLS regression to explore the impact of digital transformation. However, as a factor in enterprises' independent decisions, digital transformation has strong subjectivity, leading to strong endogeneity. To eliminate its impact, we establish a difference-in-differences model for the research that establishes dual dummy variables for the regression to lower endogeneity. There are a few previous studies on the impact of digital transformation on enterprise core competitiveness that reflect the fundamental survival factor of enterprises. At the same time, previous studies have not studied the enterprises in the industry that would benefit more from the digital transformation from the industry perspective; therefore, we build an evaluation index system of enterprise core competitiveness, analyses the role and mechanism of digital transformation affecting the core competitiveness of enterprises, and proposes policy and practical guidance related to digital transformation to promote the core competitiveness of enterprises.

We select A-share-listed steel industry enterprises as the research object. As the most traditional manufacturing industry, the steel industry has completed substantial informatization and automation infrastructure construction after completing the "Steel Industry Adjustment and Upgrade Plan (2016–2020)", with certain digital capabilities. However, the effectiveness of digital transformation is unknown; therefore, selecting the steel industry as the research object has certain practical significance.

The remaining structure of this study is as follows. The second section analyses the theoretical mechanism of digital transformation that affects the core competitiveness of traditional manufacturing enterprises and establishes a double difference model. The third and fourth sections establish an evaluation system for the core competitiveness of enterprises, showcasing preliminary regression results and their explanations. Sections 5 and 6 focus on heterogeneity analysis and robustness testing, Section 7 conducts mechanism testing, and Section 8 summarizes the conclusion, recommendations, and directions for future research.

## 2. Literature review and research hypothesis

### 2.1. Literature review

The academic community has not reached a consensus on the concept of digital transformation. The literature has indicated that scholars have generally noted that digital transformation is based on digital technology and data elements [18]. Data are regarded as important resources for enterprise value creation, and through the integration and restructuring of enterprise organizational structures, business processes, and business models[19], promoting the process of promoting deep collaboration between digital technology and the traditional production factors of enterprises [20]. Some scholars have noted that digital transformation refers to the collection of digital activities in which enterprises use digital technology to discover market opportunities and environmental changes through the application of specific technology combinations, such as data, computing, communication, and connectivity, thereby improving their competitiveness [21, 22].

The concept of enterprise core competitiveness was first proposed by Canadian scholar Stephen Herbert Hymer in 1960; however, he did not conduct a more in-depth study of it. In the 1980s, Michael Porter enriched and promoted research on enterprise competitiveness by introducing the Competition trilogy. He noted that enterprise competitiveness comes from the excess value created by products and services for customers, and an appropriate global strategy is an important guarantee for enterprises to achieve international competitiveness. After Porter, domestic and foreign scholars developed different understandings of enterprise competitiveness. For example, Prahalad and Hamel noted that proprietary technology is a special advantage for enterprises to utilize to enhance their competitiveness, and it can bring unique products and services to customers upstream and downstream of the supply chain. Although scholars have different definitions of enterprise competitiveness, they have all noted that enterprise competitiveness is a comprehensive ability that has advantages over competitors in resource integration, wealth accumulation, customer service, and other aspects.

When studying the relationship between digital transformation and core competitiveness, the academic community has mostly concentrated on performance, innovation performance, performance, and other aspects. There are four main research results, namely, positive correlation, inverted U-shaped correlation, negative correlation, and uncertainty.

**2.1.1. Positive impact of digital transformation on core competitiveness.** Mikalef et al. noted that information technology improves enterprise performance by helping enterprises plan production rationally [1], respond quickly to consumer demand, and increase organizational flexibility and agility, thereby enhancing their core competencies while optimizing internal and external communication, indirectly improving enterprise performance, and enhancing enterprise competitiveness [23]. Javier et al. noted that digital platform capabilities can have a significant positive impact on the performance of entrepreneurial SMEs through network capabilities [2]. Stefano et al. pointed out that digital technology is an effective driving force for entrepreneurship and improvements in enterprise competitiveness in the post-COVID-19 era [3]. Bertani et al. proposed that the competitiveness of digital assets can lead to

higher productivity [4]. Ferreira noted that digital transformation has become a way to gain a competitive advantage and differentiate companies and that process digitization can help improve enterprise competitiveness [24]. Digital platform capabilities can have a significant positive impact on the performance of entrepreneurial SMEs through network capabilities, thereby enhancing enterprise competitiveness [25]. Chen noted that digital transformation is crucial for enterprises to gain a competitive advantage in the digital economy era, and environmental uncertainty and resource allocation are key factors that determine the success of the transformation [11]. Some scholars have also explored the relationship between digital transformation and green innovation performance. Khan S.A. and Rehman Khan SA found that green digital technology adoption improves supply chain performance [5–7]. Additionally, Yin S found that digital technology enables green innovation in the manufacturing industry [8–10].

**2.1.2. Inverted U-shape relationship of the impact of digital transformation on core competitiveness.** KOHTAMAKI, based on the empirical analysis of financial and service digitalization data of more than 7000 credit cooperatives in the United States, found that the scope of service digitalization has an inverted U-shaped relationship with enterprise performance [15]. GEBAUER found that digital transformation is not an invariable promotion related to the core competitiveness of enterprises. Too much investment in digitalization leads to negative returns. He called this phenomenon the digital paradox [14]. Yu Feifei found an inverted U-shaped relationship between enterprise digitalization and innovation performance using empirical data on 283 enterprise digitalization questionnaires. These scholars noted that the costs and benefits of enterprise digital transformation should be considered and in line with their conditions and that blindly carrying out digital transformations may damage enterprises' core competitiveness.

**2.1.3. Negative impact of digital transformation on core competitiveness.** Some studies have found a negative relationship between digital transformation and enterprise competitiveness or that their relationship is not significant. Some scholars have noted that enterprises incur many learning and management costs in the digital transformation process that to some extent hinders the effectiveness of digital transformation and harms enterprise competitiveness [12, 13]. At the same time, more than half of the companies that adopt digital transformation strategies have poorer performance than before, and some may face the risk of bankruptcy [26].

**2.1.4. Digital transformation has no impact on core competitiveness.** The research results of Kim showed no direct positive correlation between digital technology and enterprise performance [17]. Hajli et al. found that only some enterprises benefit from digitization, while others do not, such as the banking industry in Nigeria and the United Kingdom. This phenomenon is known as the "IT paradox" [16].

## 2.2. Research hypothesis

This part mainly puts forward the core hypothesis of this article based on the theory and puts forward the hypothesis on the process and adjustment mechanism of Digital transformation affecting core competitiveness. Fig 1 shows the theoretical framework of this article.

**2.2.1. Digital transformation and core competitiveness of enterprises.** According to resource-based theory, the core of an enterprise's competitive advantage lies in how to obtain heterogeneous resources, that is, resources with high value that are scarce, difficult to copy, and irreplaceable. Digital transformation, as a unique heterogeneous resource of enterprises, can help them improve their core competitiveness and thus gain a competitive advantage.

Digital transformation refers to the process of taking digital products and technologies as the fulcrum to encourage enterprises to change at multiple levels [27]. In this process,

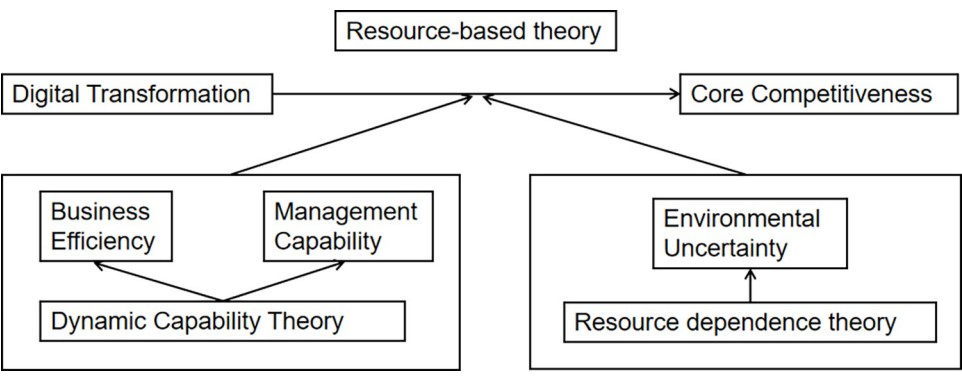

**Fig 1. Hypothesis framework.**

enterprises need new technologies, ecological positioning, business models, business and organizational processes, and intelligent soft power to achieve more effective competition in the rapidly changing digital economy [28]. According to the theory of IT capability and resource-based theory, as a unique resource input, digital transformation can help enterprises gain competitive advantages, thus improving their core competitiveness. In the market environment of VUCA, through digital technology, manufacturing enterprises can realize the supervision and correction of the whole process–from research and development to production–and realize the efficient use of resources, thus reducing production costs and increasing profits [29]. Second, the surge in computing power has enabled people to use machine learning to solve increasingly complex problems, and the rapid growth in the convenience of data acquisition and storage has also enabled machine learning to be applied in more fields [30]. Third, at the business level, the platform brought by digital technology can integrate all stakeholders in the "R&D—raw materials—production—sales—after-sales" chain, making it is easier to obtain feedback at all levels and reflect it in these nodes, realize the optimization of each node, perfect the chain, realize the association of all stakeholders [31], and achieve the efficient cooperation among enterprises. Given this, we propose Hypothesis H1.

H1: Digital transformation can promote the core competitiveness of enterprises.

At the same time, when business processes change through digital transformation, the internal organizational structure, resource allocation, scheduling management, and other aspects of the enterprise need to be transformed accordingly [32]. Enterprise employees also need to have the skills to adapt to digital transformation. These factors may hinder or delay the impact of digital transformation, which indicates that digital transformation produces stable results. Instead, the series of optimizations should be continuous [33]. Kane also noted that organizations cannot instantly and skillfully apply digitalization [34]. The digital transformation phenomenon is a gradual process that unfolds throughout the organization over time, continuously promoting the integration of enterprise management systems and digital technology and constantly and repeatedly debugging business processes, organizational structures, resource allocation, technical training, etc. This debugging process represents business and management digitalization. Continuous coupling and mutual adaptation of technology digitization achieve results that improve input–output efficiency. Therefore, since the application of the digital transformation results cannot be reflected synchronously in business performance, the digital transformation conducted in the first year may require a second year or even longer for the results to be reflected. Therefore, this article proposes Hypothesis H2 based on Hypothesis H1.

H2: There is a lag effect of digital transformation on improvements in core competitiveness.

**2.2.2. The mechanism of digital transformation to promote enterprise core competitiveness.** *1.Internal mechanism.* According to the theory of dynamic capabilities, dynamic capabilities refer to the continuous integration and reconfiguration of resources–updating and recreating resources and capabilities [35]. In the process of enhancing core competitiveness through digital transformations, the level of the dynamic capabilities of an enterprise determines the efficiency and effectiveness of the improvements. High levels of dynamic capabilities cannot ensure that all of an enterprise's strategic decisions are accurate; however, these capabilities can enable enterprises to respond effectively when encountering accidents and errors, and dynamic capabilities play a positive role in improving enterprise competitiveness and performance.

As two dynamic capabilities of an enterprise, enterprise management capability and enterprise operational efficiency may play a catalytic role in promoting its core competitiveness through digital transformations.

During digital transformations, digital strategy and digital technology are equally important components [36]. Digital strategy reflects the integration of the digital development goals and digital resources of enterprises, while digital technology reflects the digital methods of enterprises' organizational management and production models. Some scholars have found that enterprises can incur many learning and management costs during the digital transformation process, which to some extent hinders the effectiveness of digital transformation and damages enterprises' competitiveness [12, 13]. Therefore, relying solely on technology cannot ensure the full achievement of digital transformation; instead, enterprises need to develop reasonable digital transformation strategies that require qualified management skills. In addition, an enterprise's operational efficiency reflects its resource allocation ability to a certain extent. During the digital transformation process, the stronger the resource allocation ability is, the stronger is the ability of the generated digital technologies to collaborate [20], further promoting an improvement in the enterprise's core competitiveness. Therefore, this article proposes Hypothesis H3.

H3: Enterprises' management abilities and business efficiency play roles in promoting the impact of digital transformation on their core competitiveness.

*2.External mechanism.* In a highly uncertain environment, competition among enterprises is intensified. If enterprises can quickly adapt to environmental changes and realize digital transformations in a complex environment, they can enhance their competitiveness. According to growth option theory, for enterprises committed to digital transformations, a highly uncertain environment not only does not constitute an external threat in the enterprise development process but can also help give enterprises opportunities to improve their competitive advantages and narrow their gaps with peers. When environmental uncertainty is high, market demand diversifies. Enterprises engaged in digital transformation increase their investments in innovation and develop new-generation products to identify potential market opportunities. In addition, in a highly volatile environment, a large number of competitors rush into the market, which reduces enterprises' market shares and profitability, thus enhancing their competition. A powerful means for enterprises to avoid being eliminated by the market is to strengthen their core competitiveness. Therefore, we propose Hypothesis H4.

H4: Environmental uncertainty plays a catalytic role in the impact of digital transformation on the core competitiveness of enterprises.

## 3. Research design

### 3.1. Data source

In this paper, we select steel industry enterprises listed on the Shanghai and Shenzhen A-shares markets from 2010 to 2020 as the research object and conduct the following processing on the

**Table 1. Enterprise core competitiveness index system.**

| Indicator type | Indicator name | data sources |
|---|---|---|
| Enterprise profitability | Cost rate | Total cost/operating income |
| | Earnings per share | Current net profit attributable to ordinary shareholders/weighted average number of ordinary shares issued in the current period |
| | Return on net assets | Net profit/average net assets |
| Enterprise growth ability | Operating profit growth rate | Increase in operating profit of current year/total operating profit of previous year |
| | Net profit growth rate | Net profit growth of this year/net profit of last year |
| | Total asset growth rate | Total asset growth of this year/total assets of last year |
| Enterprise operation ability | Total asset turnover | Total sales revenue/average total assets |
| | Asset-liability ratio | Total liabilities/total assets |
| Employee value | The relative value of employees | Cash paid to and for employees/total number of employees |
| | Per capita salary of employees | Payroll payable/total number of employees |
| | Proportion of R&D personnel | Number of R&D personnel/total number of employees |
| Innovation ability | Innovation investment | R&D expenditure |
| | Intangible asset ratio | Intangible assets/total assets |

entire sample. First, ST samples and samples delisted during the period are eliminated. Second, samples with missing variable data for more than two consecutive years are eliminated. Third, to reduce the impact of outliers, we conduct 1% and 99% tail reductions for all continuous variables at the micro level. The raw data are from CSMAR and RESSET database. See Table 1 for the descriptive statistics of the variables.

## 3.2. Variable setting

**3.2.1. Explained variable.** We select the core competitiveness of enterprises as the explanatory variable. Many scholars have researched the measurement of the core competitiveness of enterprises. The general methods include the analytic hierarchy process, principal component analysis, and text analysis. Based on the research of Chen Y [37], we adopt the principal component analysis method to comprehensively measure the core competitiveness of enterprises from the perspectives of enterprise profitability, enterprise growth, enterprise operation, human resources, and innovation ability. Compared to traditional financial indicators that ignore the impact of employee quality and ability on the core competitiveness of enterprises, we add the perspective of human resources to more comprehensively measure the core competitiveness of enterprises.

**3.2.2. Control variables.** To increase the accuracy of the research, a series of control variables are added in this paper, including enterprise age (Age), enterprise size (Size, natural logarithm one plus total assets), enterprise revenue scale (Sale, the logarithmic treatment of operating income), cash flow intensity (Cash, ratio of cash and its cash equivalents to total assets), equity concentration (S-D, concentration of top ten shareholders), and the combination of two positions (Dual, if the chairperson and general manager is the same person, then the value is 1 and 0 otherwise), Audit opinion (Audit, if the standard unqualified opinion is issued by the accounting firm then the value is 0 and 1 otherwise). Table 2 provides a description of the above statistics.

**Table 2. Descriptive statistics.**

| Variable name | Observations | Average | SD | Min | Max |
|---|---|---|---|---|---|
| CORE | 578 | -0.01 | 0.74 | -3.175 | 2.27 |
| did | 578 | 0.213 | 0.41 | 0 | 1 |
| Age | 593 | 13.924 | 5.185 | 2 | 27 |
| SD | 595 | 64.661 | 14.001 | 23.538 | 95.094 |
| Dual | 595 | 0.118 | 0.322 | 0 | 1 |
| Audit | 596 | 0.971 | 0.167 | 0 | 1 |
| Sale | 548 | 23.102 | 1.69 | 17.814 | 26.624 |
| Size | 552 | 23.28 | 1.548 | 19.318 | 26.664 |
| Cash | 547 | 0.979 | 0.64 | 0.029 | 5.187 |

## 3.3. Model settings

Ashenfelter (1976) were the first to propose the use of DID methods for the assessment of the public policy effect [38]. Since then, there has been a proliferation of research results on DID methods. Zhang, Q (2022) used the DID model and discovered that green innovation output increases significantly through the implementation of corporate digital transformation [39]. Wang, Q (2022) found that digital transformation significantly reduces electricity consumption and intensity, and this electricity-saving effect is achieved through technological optimization and industrial upgrading brought about by digital transformation [40]. Tao Zhang (2021) revealed that the implementation of digital transformation plays a significant role in promoting economic benefits [41].

There are many more management papers that have used the DID method to research digital transformation, which indicates that DID is suitable for researching this topic.

Therefore, to verify Hypothesis H1, we set the following double difference model:

The core explanatory variable of this model is the interaction item of treat and time. The treat variable is used to delineate the experimental group and the control group. We obtain the degree of digital transformation of the steel industry from the CSMAR database and perform logarithmic processing. During the observation period, if the degree of digitalization is greater than the average value of the sample, the variable takes the value of 1 and 0 otherwise. The time variable is used to divide the time point of the impact of digital transformation. Considering that the Guiding Opinions of the State Council on Actively Promoting the Action of "Internet plus" issued by the State Council in 2015 initiated the development pattern of China's digital economy, we select 2015 as the policy time point. After that year, the value of this variable is 1 and 0 before that year.

$$Core_{it} = \alpha_0 + \alpha_1 treat_i \times time_t + \sum \alpha_2 X_{it} + Year_t + Id_i + \varepsilon_{i\tau} \tag{1}$$

To verify Hypotheses H3 and H4, we set the following double difference model:

$$Core_{it} = \alpha_0 + \alpha_1 treat_i \times time_t + \alpha_2 tj + \alpha_3 treat_i \times time_t \times tj + \sum \alpha_4 X_{it} + Year_t + Id_i + \varepsilon_{i\tau} \tag{2}$$

where i represents the individual enterprise, t represents the time effect, Core represents the core competitiveness of the enterprise, and treat*time is the core explained variable of this article, treat is the policy grouping variable (1 for the experimental group, 0 for the control group), time is the policy time point variable (that takes the 1 after 2015 and 0 otherwise), Xit is a series of enterprise-level control variables, Year and Id are the fixed effects of the year and individual, ε it is the error disturbance item, and Tj is the regulating variable.

**Table 3. KMO inspection and Bartlett spherical inspection.**

| Determinant of the correlation matrix | |
|---|---|
| DET | 0.230 |
| Bartlett test of sphericity | |
| Chi-square | 837.946 |
| Degrees of freedom | 78 |
| p value | 0.000 |
| H0: variables are not intercorrelated | |
| Kaiser–Meyer–Olkin Measure of Sampling Adequacy | |
| KMO | 0.593 |

## 4. Benchmark regression results

### 4.1. Measurement of core competitiveness

**4.1.1. Feasibility analysis.** After the typical normalization of the selected variables and sample data were conducted, the KMO and Bartlett spherical tests were carried out. The test results are shown in the following Table 3. The measurement value of KMO is 0.593, which is close to 0.6, indicating that the partial correlation between the selected variables is sufficiently strong. The P value of the Bartlett spherical test is 0.000, indicating that the selected variables and the data passed this test, and there is a correlation between the variables that can be used for further principal component analysis.

**4.1.2. Core competitiveness score.** The regression method is used to calculate the score of common factors and obtain the component matrix diagram shown in the Table 4. The component matrix shows the cumulative variance contribution rate of the six extracted principal component factors. According to the component score coefficient matrix, the expression of each principal component factor and the final core competitiveness score can be determined.

According to the above table, six principal components with a characteristic value greater than 1 are selected, and the cumulative contribution rate reaches 63%. The first six factors can reflect most of data levels, while the characteristic value is less than 1 and gradually decreases from the sixth factor, which has little effect on the interpretation and substitution of the original variables.

**Table 4. Cumulative variance contribution rate of principal component factors.**

| Component | Eigenvalue | Difference | Proportion | Cumulative |
|---|---|---|---|---|
| F1 | 2.29259 | 0.76285 | 0.1764 | 0.1764 |
| F2 | 1.52974 | 0.27503 | 0.1177 | 0.2940 |
| F3 | 1.25471 | 0.12841 | 0.0965 | 0.3905 |
| F4 | 1.12630 | 0.04933 | 0.0866 | 0.4772 |
| F5 | 1.07697 | 0.06947 | 0.0828 | 0.5600 |
| F6 | 1.00751 | 0.03677 | 0.0775 | 0.6375 |
| F7 | 0.97073 | 0.10082 | 0.0747 | 0.7122 |
| F8 | 0.86991 | 0.11858 | 0.0669 | 0.7791 |
| F9 | 0.75133 | 0.07647 | 0.0578 | 0.8369 |
| F10 | 0.67486 | 0.06632 | 0.0519 | 0.8888 |
| F11 | 0.60855 | 0.08706 | 0.0468 | 0.9356 |
| F12 | 0.52148 | 0.20618 | 0.0401 | 0.9757 |
| F13 | 0.31531 | 0.10768 | 0.0243 | 1.00 |

The calculation formula of the core competitiveness score is

$$SCORE = (0.1764 \times F1 + 0.1177 \times F2 + 0.0965 \times F3 + 0.0866 \times F4 + 0.0828 \times F5 + 0.0775 \times F6)/0.6375$$

According to the calculated score, as the representative variable of the enterprise's core competitiveness, the higher the score is, the stronger is the enterprise's core competitiveness.

## 4.2. Parallel trend test

Before conducting the double-difference empirical analysis, a parallel trend test chart is drawn to test whether the experimental and control groups meet the parallel trend hypothesis. As shown in Fig 2, the relative time of the policy implementation is treate5, which indicates the year of the enterprise's digital transformation. Before the implementation of digital transformation, the regression coefficient of the policy effect is not significantly different from 0, indicating that there is no significant difference between the changing trend of enterprises in the experimental and control groups before the implementation of digital transformation.

## 4.3. Benchmark regression results

In the benchmark regression, we adopt a progressive regression strategy. Table 5 shows the regression results. Model (1) only controls the fixed effects of time and industry, and the regression coefficient of DID is 0.276 and passes the statistical significance test of 1%. In Model (3), the control variable set is included on the original basis, and the relevant regression coefficient is reduced (0.213), which may be caused by the absorption of some factors that affect the core competitiveness of enterprises after the control variable is included; however, it

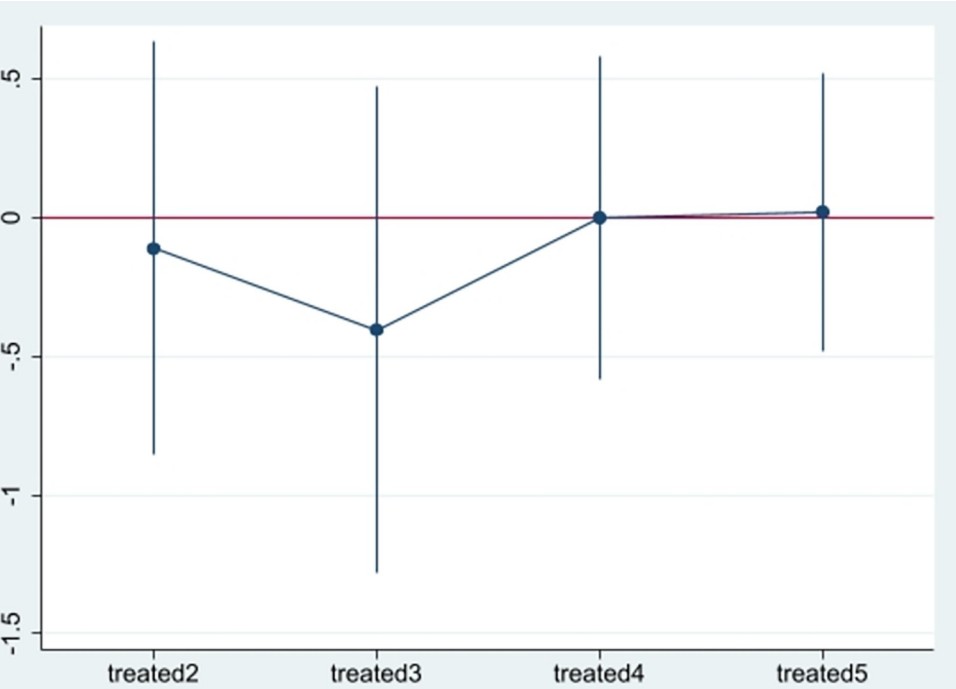

**Fig 2. Parallel trend test.**

**Table 5. Benchmark regression results.**

| VARIABLES | (1) CORE | (2) CORE | (3) CORE | (4) CORE |
|---|---|---|---|---|
| did | 0.276*** | | 0.213** | |
| | (3.09) | | (2.42) | |
| L.did | | 0.266*** | | 0.253** |
| | | (2.59) | | (2.50) |
| Age | | | 0.012 | 0.013 |
| | | | (1.48) | (1.51) |
| SD | | | 0.003 | 0.005 |
| | | | (1.12) | (1.55) |
| Dual | | | 0.155 | 0.120 |
| | | | (1.48) | (1.08) |
| Audit | | | 0.233 | 0.305* |
| | | | (1.42) | (1.74) |
| Sale | | | 0.637*** | 0.717*** |
| | | | (4.69) | (4.91) |
| Size | | | -0.455*** | -0.557*** |
| | | | (-3.09) | (-3.48) |
| Cash | | | 0.066 | 0.006 |
| | | | (0.46) | (0.04) |
| Constant | 0.067 | -0.025 | -4.796*** | -4.528** |
| | (0.80) | (-0.29) | (-2.72) | (-2.48) |
| Observations | 578 | 529 | 547 | 507 |
| R-squared | 0.217 | 0.221 | 0.295 | 0.309 |
| Number of Ids | 51 | 51 | 51 | 51 |
| Company FE | YES | YES | YES | YES |
| Year FE | YES | YES | YES | YES |

z-statistics in parentheses

*** $p<0.01$

** $p<0.05$

* $p<0.1$

is still significant at below the 5% level. This indicates that the higher the degree of digitalization of enterprises is, the greater is the core competitiveness of traditional manufacturing enterprises. There is a significant positive correlation between the two. Therefore, Hypothesis H1 of this paper is supported by empirical evidence. Both Model (2) and Model (4) lag the core explanatory variable by one period, and the result is still significant. Therefore, Hypothesis H2 in this paper is supported by empirical evidence, and the coefficient (0.253) of Model (4) is larger than that of Model (3), which indicates that the effect of the previous period of digitalization on the core competitiveness of the next period is more obvious, forming a promoting role with superimposed characteristics. This has stimulated a rise in the core competitiveness of enterprises to a greater extent, possibly partly because digital transformation as a process may not be completed in the current period or has not yet been applied to the specific business of enterprises, which makes its impact on the core competitiveness of enterprises partially lagging and providing additional evidence for core research Hypothesis H2 of this paper.

# 5. Heterogeneity test

## 5.1. Quantile regression

Even the unified external environment has different impacts on different enterprises, let alone the digital transformation carried out independently by enterprises. Therefore, for enterprises with core competitiveness in different industry positions, because the promotional effect of digital transformation on enterprise core competitiveness may be different, we introduce quantile regression to explore this issue.

Table 6 shows the results of the quantile regression. The effect of digital transformation is more significant for enterprises in the 20% and 80% quantile positions of core competitiveness, and the DID coefficient is also higher. This shows that the effect of digital transformation is more obvious for enterprises that are backwards and leading in the industry, while the effect of digital transformation is not significant for enterprises in the middle reaches. This finding is also in line with the conclusion of Hajli that not all enterprises benefit from digital transformation [16].

Fig 3 reports the distribution track of CORE, which conforms to the normal distribution; that is, the number of enterprises on both sides is large, and the number of enterprises on both sides is small. If the number of intermediate enterprises is large enough and the number of enterprises on both sides is small enough, the conclusion of scholars' research is likely to be that digital transformation has no or even a negative impact on enterprises, which also confirms why there is an "IT paradox" or "digital paradox" and why some scholars come to the counterintuitive conclusion that digital transformation has no or even a negative impact on enterprise performance.

## 5.2. Nature of property rights

Compared with nonstate-owned enterprises, state-owned enterprises have key advantages when constructing new infrastructure, such as artificial intelligence, cloud computing, and Internet of Things. In September 2020, the State-owned Assets Supervision and Administration Commission issued the Notice on Accelerating the Digital Transformation of State-owned Enterprises. Governments at all levels actively promoted the digital transformation of state-owned enterprises, and these enterprises responded positively, playing a leading role in

**Table 6. Quantile regression results.**

|  | (1) | (2) | (3) | (4) |
|---|---|---|---|---|
|  | **20%** | **40%** | **60%** | **80%** |
| time | 0.048 | 0.170*** | 0.299*** | 0.456*** |
|  | (0.55) | (2.64) | (5.69) | (4.63) |
| treat | -0.619*** | -0.002 | -0.104 | -0.138 |
|  | (-3.15) | (-0.01) | (-0.84) | (-0.62) |
| did | 0.818*** | 0.147 | 0.183 | 0.567** |
|  | (3.52) | (0.83) | (1.27) | (2.14) |
| _cons | -0.502*** | -0.217*** | -0.020 | 0.231*** |
|  | (-8.37) | (-4.88) | (-0.55) | (3.38) |
| N | 578 | 578 | 578 | 578 |

t statistics in parentheses

* $p < 0.1$

** $p < 0.05$

*** $p < 0.01$

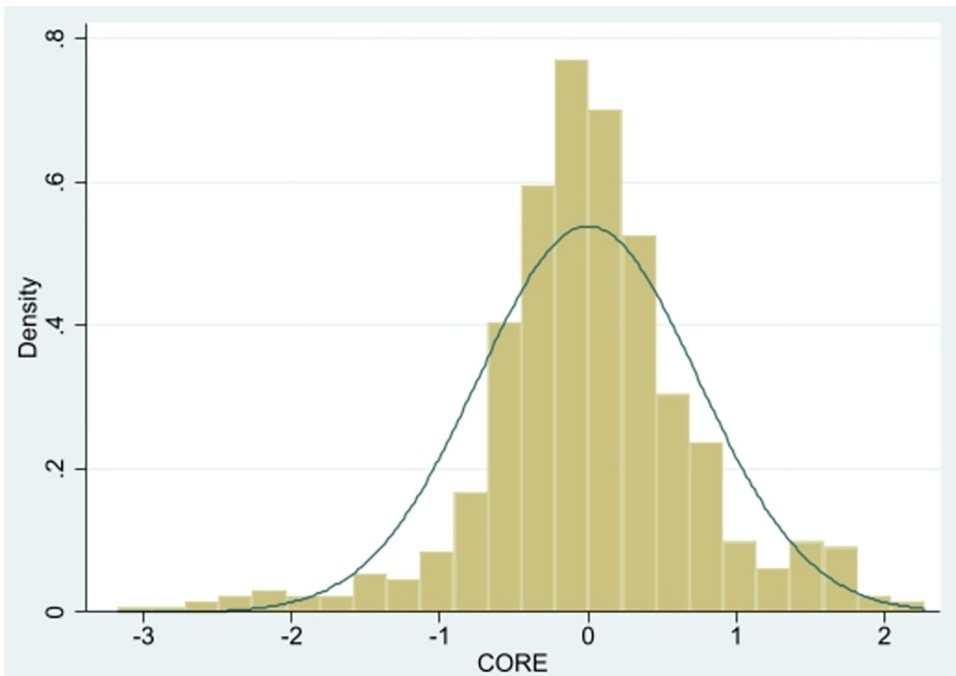

**Fig 3. CORE normal distribution.**

the digital technological revolution and industrial transformation Therefore, we note that the digital transformation of state-owned enterprises is more effective in empowering enterprises' core competitiveness.

The regression results are reported in Columns (1) and (2) of Table 7. The results show that compared with nonstate-owned enterprises, the digital transformation of state-owned enterprises has a more obvious effect on promoting their core competitiveness.

### 5.3. Government subsidies

Government subsidies are undoubtedly a tonic for enterprises. Government subsidies have played a catalytic role in promoting the development of enterprises in all aspects. Therefore, we believe that the digital transformation of enterprises with high government subsidies has a more obvious effect on promoting their core competitiveness. In this paper, the average value of government subsidies is taken as the zero point. If this value is higher than the average value, it is regarded as a high government subsidy group, and if it is lower than the average value, it is regarded as a low government subsidy group.

The regression results are reported in Columns (3) and (4) of Table 7. The results show that the digital transformation of enterprises with high government subsidies (coefficient 0.331) has a more obvious effect on their core competitiveness than that of enterprises with low government subsidies (coefficient 0.271).

## 6. Robustness check

### 6.1. PSM–DID

To eliminate the systematic difference in the changing trend between the experimental group and the control group, the propensity score matching–double difference method (PSM–DID) was used for the robustness test. The estimated results are listed in Table 8. After adding a

**Table 7. Property rights and government subsidies.**

| VARIABLES | (1) CORE State-owned enterprises | (2) CORE Non State-owned enterprises | (3) CORE High government subsidies | (4) CORE Low government subsidies |
|---|---|---|---|---|
| did | 0.403*** | 0.001 | 0.331** | 0.271** |
|  | (3.88) | (0.01) | (2.30) | (2.19) |
| Age | -0.001 | 0.013 | 0.004 | 0.005 |
|  | (-0.06) | (0.91) | (0.25) | (0.52) |
| SD | -0.001 | 0.003 | -0.008 | 0.003 |
|  | (-0.13) | (0.54) | (-1.11) | (0.73) |
| Audit | 0.663*** | -0.137 | -0.061 | 0.433** |
|  | (3.28) | (-0.51) | (-0.18) | (2.15) |
| Sale | 0.203 | 1.010*** | 1.185*** | 0.362** |
|  | (1.12) | (2.93) | (4.65) | (2.10) |
| Size | 0.236 | -1.024*** | -0.528* | -0.347* |
|  | (1.20) | (-3.19) | (-1.95) | (-1.78) |
| Cash | 0.733*** | -0.485 | -0.197 | 0.286 |
|  | (3.75) | (-1.38) | (-0.92) | (1.40) |
| Dual | 0.431** | 0.087 | 0.392** | 0.192 |
|  | (1.96) | (0.60) | (1.98) | (1.31) |
| Constant | -10.827*** | 0.640 | -13.479** | -1.363 |
|  | (-5.44) | (0.16) | (-2.43) | (-0.63) |
| R2 | 0.450 | 0.282 | 0.439 | 0.282 |
| Observations | 313 | 182 | 189 | 358 |

z-statistics in parentheses

*** p<0.01

** p<0.05

*p<0.1

fixed effect and control variable regression, the coefficient decreased; however, the result was still stable at the 0.1 level.

## 6.2. Placebo test

To test whether the digital transformation has the growth effect brought about by time changes and to exclude the impact of unobserved corporate sample characteristics on the regression results, 123 samples were randomly selected from all 549 samples as a "pseudo experimental group" for placebo testing. The random sampling process was repeated 500 times, and the product of the random sampling process and the time dummy variable were used as the core explanatory variable regressionsion. Fig 4 shows the coefficient distribution of the regression results and that the distribution of the regression coefficients is concentrated around 0, indicating that the sample combination after random sampling has no impact on the core competitiveness of the enterprise. Therefore, the regression results of the benchmark regression that distinguish the experimental and control group through participation are robust.

## 6.3. Replace the interpreted variable

We replaced and retested the indicators used to measure the core competitiveness of enterprises. 1. According to the research of Monte, the core competitiveness of enterprises is

**Table 8. PSM–DID estimation results.**

| VARIABLES | (1) CORE | (2) CORE | (3) CORE | (4) CORE |
|---|---|---|---|---|
| did | 0.389*** | 0.215** | 0.338*** | 0.145* |
|  | (4.64) | (2.40) | (4.12) | (1.64) |
| Age |  |  | 0.019*** | 0.012 |
|  |  |  | (3.11) | (1.56) |
| SD |  |  | 0.001 | 0.004 |
|  |  |  | (0.42) | (1.29) |
| Dual |  |  | 0.218** | 0.086 |
|  |  |  | (2.10) | (0.79) |
| Audit |  |  | 0.057 | 0.083 |
|  |  |  | (0.34) | (0.51) |
| Sale |  |  | 0.522*** | 0.826*** |
|  |  |  | (4.43) | (5.27) |
| Size |  |  | -0.572*** | -0.588*** |
|  |  |  | (-4.75) | (-3.76) |
| Cash |  |  | 0.079 | -0.069 |
|  |  |  | (0.74) | (-0.47) |
| Constant | -0.019 | 0.084 | 0.737 | -5.819*** |
|  | (-0.28) | (1.03) | (1.00) | (-3.16) |
| Observations | 549 | 549 | 518 | 518 |
| R-squared | 0.0453 | 0.190 | 0.1638 | 0.273 |
| Number of Ids | 51 | 51 | 51 | 51 |
| Company FE | NO | YES | NO | YES |
| Year FE | NO | YES | NO | YES |

z-statistics in parentheses

*** p<0.01

** p<0.05

* p<0.1

measured by the average sales growth rate. 2. Since the return on net assets reflects the ability of an enterprise to profit from its assets, it can to some extent reflect the core competitiveness of the enterprise. Therefore, the return on net assets measures the competitiveness of the enterprise. Table 9 reports the regression results, which are still significant.

## 7. Adjustment mechanism test

### 7.1. Mechanism analysis: Enterprise management capability

In this paper, enterprise organizational capital is used to represent enterprise management capabilities. Based on the research by Eisfeldt et al., the perpetual inventory method is used to measure organizational capital [42]. The cumulative sales, general, and administrative (SG&A) expenses that measure the enterprise's organizational capital stock are used to represent enterprise management capabilities; that is, the organizational capital (OC) in SG&A expenditures is used to represent enterprise management capabilities. The definition of SG&A expenses in United States GAAP refers to all commercial operating expenses (i.e., expenses not directly related to product production) incurred by a company in the ordinary course of business transactions. We use the perpetual inventory method to construct OC through the specific steps as follows.

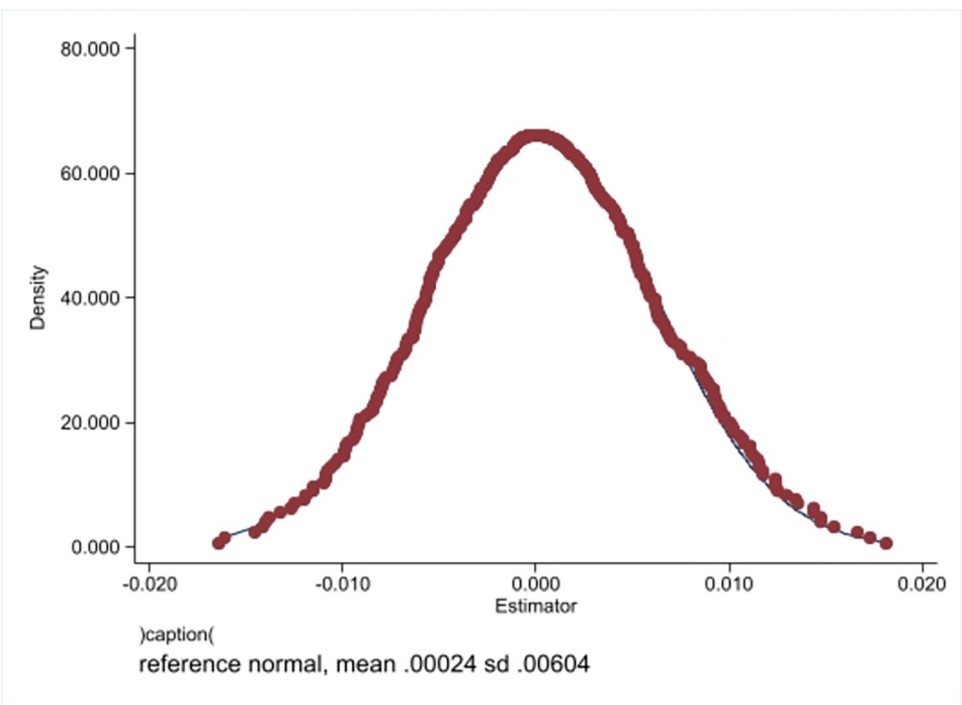

**Fig 4. Placebo test.**

The first step is to calculate the initial value. The formula is as follows:

$$OC_0 = \frac{SG\&A_1}{g + \delta_0}$$

where is the organizational capital of the initial year; the discount rate of organizational capital is generally 15%; $SG\&A_1$ is the sum of sales and administrative expenses in the next year of the initial year; and is assigned the value of 10% according to Zhang Tijun (2022). The second step is to calculate the value of the remaining years using the formula:

$$OC_{i,t} = (1 - \delta_0)OC_{i,t-1} + \frac{SG\&A_{i,t}}{CPI_t}$$

where i is the enterprise, t is the year, and CPI is the consumer price index.

The coefficient of DID in the benchmark regression model is positive, and the coefficient of did#OC in the adjustment effect model is still positive, indicating that the enterprise's management ability plays a promoting role in improving the core competitiveness of digital transformation. The result is still significant after controlling for the fixed effect and adding the control variable set.

## 7.2. Mechanism analysis: Environmental uncertainty

According to the method in Shen Huihui, business income data from the enterprise's first five years and the old method are used to construct the environmental uncertainty index (EU) [43].

$$revenue = \varphi_0 + \varphi_1 Year + \varepsilon$$

Revenue represents sales revenue, and Year represents the annual variable. If the observation value is the fourth year in the past, then Year = 1; if the observation value is the third year in the

**Table 9. Substitution of interpreted variables.**

| VARIABLES | (1) ROA | (2) ROA | (3) Sales growth rate | (4) Sales growth rate |
|---|---|---|---|---|
| did | 3.552*** | 1.845*** | 1.370*** | 0.619** |
|  | (5.77) | (2.61) | (5.18) | (2.13) |
| Age | 0.238*** | 0.165** | 0.146*** | 0.094*** |
|  | (4.02) | (2.30) | (5.38) | (2.99) |
| SD | 0.002 | 0.020 | 0.007 | 0.006 |
|  | (0.09) | (0.74) | (0.64) | (0.57) |
| Dual | -0.336 | -0.888 | 0.126 | 0.098 |
|  | (-0.37) | (-0.93) | (0.31) | (0.25) |
| Audit | 0.640 | 0.489 | 0.982 | 0.869 |
|  | (0.41) | (0.32) | (1.64) | (1.55) |
| Sale | 3.431*** | 5.752*** | 1.079** | 1.065* |
|  | (3.51) | (4.57) | (2.22) | (1.89) |
| Size | -4.588*** | -5.172*** | -0.399 | -0.391 |
|  | (-4.45) | (-3.80) | (-0.78) | (-0.65) |
| Cash | 0.242 | -0.311 | 0.041 | 0.271 |
|  | (0.25) | (-0.24) | (0.08) | (0.43) |
| Constant | 26.617*** | -11.887 | -17.911*** | -16.940** |
|  | (3.75) | (-0.74) | (-4.62) | (-2.56) |
| Observations | 583 | 583 | 493 | 493 |
| R-squared | 0.177 | 0.288 | 0.232 | 0.365 |
| Number of Ids | 51 | 51 | 50 | 50 |
| Company FE | NO | YES | NO | YES |
| Year FE | NO | YES | NO | YES |

z-statistics in parentheses

*** p<0.01

** p<0.05

* p<0.1

past, then Year = 2; by analogy, if the observation value is the current year, then Year = 5. The residual of the model is abnormal sales revenue. The standard deviation of the company's abnormal sales revenue in the past five years is calculated and then divided by the average value of the sales revenue in the past five years to obtain the environmental uncertainty without industry adjustment. Because the research object of this paper is a single industry, the final environmental uncertainty without industry adjustment is the actual application data of this paper.

The interaction of the EU and the core explanatory variable is added to the regression, and the results are shown in Table 10. The core explanatory variable did#EU is still significantly positive after controlling for a series of control variables and absorbing individual and time-fixed effects, indicating that environmental uncertainty strengthens the promotion of digital transformation on the core competitiveness of enterprises.

## 7.3. Mechanism analysis: Enterprise operation efficiency

Referring to the research of Chiou, the ratio of the enterprise's operating income to total assets is used to represent its operating efficiency [44].

The regression results are shown in Table 11. After controlling for a series of control variables and absorbing individual and time-fixed effects, the core explanatory variable

**Table 10. Regulatory role of management capacity.**

| VARIABLES | (1) | (2) | (3) | (4) |
|---|---|---|---|---|
| | y | y | y | y |
| did | -0.037 | -2.115 | -2.269* | -2.449* |
| | (-0.03) | (-1.45) | (-1.81) | (-1.75) |
| OC | -0.139** | -0.152** | -0.247*** | -0.047 |
| | (-2.51) | (-2.03) | (-4.12) | (-0.54) |
| did#OC | 0.098 | 0.491* | 0.513** | 0.537** |
| | (0.37) | (1.68) | (2.05) | (1.91) |
| Age | | | 0.004 | 0.010 |
| | | | (0.63) | (1.28) |
| SD | | | 0.003 | 0.003 |
| | | | (1.14) | (0.92) |
| Dual | | | 0.117 | 0.171 |
| | | | (1.06) | (1.37) |
| Audit | | | 0.252 | 0.163 |
| | | | (1.45) | (0.94) |
| Sale | | | 0.619*** | 0.662*** |
| | | | (3.81) | (3.55) |
| Size | | | -0.677*** | -0.453** |
| | | | (-3.99) | (-2.31) |
| Cash | | | 0.068 | 0.159 |
| | | | (0.36) | (0.81) |
| Constant | 0.633** | 0.947** | 2.158** | -5.232** |
| | (1.99) | (2.20) | (2.27) | (-2.49) |
| Observations | 480 | 480 | 467 | 467 |
| R-squared | 0.098 | 0.252 | 0.264 | 0.346 |
| Number of Ids | 42 | 42 | 41 | 41 |
| Company FE | NO | YES | NO | YES |
| Year FE | NO | YES | NO | YES |

z-statistics in parentheses

*** p<0.01

** p<0.05

* p<0.1

did#Efficiency is still significant at the 1% level, and the coefficient is positive, indicating that the higher the business efficiency of the enterprise is, the more obvious is the role of digital transformation in promoting its core competitiveness (Table 12).

## 8. Conclusion and implications

In recent years, given the increasing importance of digital economy development, enterprises' digital transformation has been deeply engraved in the evolution of traditional industries. This new "entity enterprise+digital" model has formed a significant potential driving force for China's innovation-driven development strategy. Given the gradual development of technology, improving the core competitiveness of enterprises through digital transformation has become one of the core means for enterprises to win in market competitions. To explore the mechanism of this process, we used the principal component analysis method and the double difference model to study the impact of digital transformation on the core competitiveness of

**Table 11. Regulation effect of environmental uncertainty.**

| VARIABLES | (1) y | (2) y | (3) y | (4) y |
|---|---|---|---|---|
| did | 1.172*** | 0.818*** | 1.072*** | 0.610*** |
| | (4.98) | (3.59) | (4.69) | (2.61) |
| EU | -0.288** | -2.668*** | -0.129 | -0.186 |
| | (-2.08) | (-5.85) | (-1.09) | (-0.25) |
| did#EU | 0.595*** | 0.497*** | 0.560*** | 0.377** |
| | (3.57) | (3.17) | (3.51) | (2.36) |
| Age | | | 0.017*** | 0.011 |
| | | | (2.71) | (1.35) |
| SD | | | 0.001 | 0.003 |
| | | | (0.23) | (0.83) |
| Dual | | | 0.191* | 0.138 |
| | | | (1.95) | (1.32) |
| Audit | | | 0.303* | 0.276* |
| | | | (1.82) | (1.68) |
| Sale | | | 0.438*** | 0.603*** |
| | | | (4.17) | (4.40) |
| Size | | | -0.491*** | -0.452*** |
| | | | (-4.40) | (-3.07) |
| Cash | | | 0.146 | 0.084 |
| | | | (1.42) | (0.59) |
| Constant | -0.479** | -3.661*** | 0.337 | -4.086** |
| | (-2.18) | (-5.36) | (0.40) | (-2.49) |
| Observations | 578 | 578 | 547 | 547 |
| Number of Ids | 51 | 51 | 51 | 51 |
| R-squared | 0.077 | 0.223 | 0.212 | 0.297 |
| Controls | NO | NO | YES | YES |
| Company FE | NO | YES | NO | YES |
| Year FE | NO | YES | NO | YES |

z-statistics in parentheses

*** $p < 0.01$

** $p < 0.05$

* $p < 0.1$

enterprises and the adjustment mechanism in the process. Previous researchers have mainly focused on innovation and corporate performance and have neglected the core competitiveness of enterprises, which is a key factor for their survival in the market. The research results are as follows.

(i) Enterprises' digital transformation has significantly improved their core competitiveness. For the lagger and the leader in the industry, the effect of digital transformation is more obvious. (ii) Digital transformation has a certain time lag effect on improvements in the core competitiveness of enterprises. The effect of the previous period's digital transformation on the next period's core competitiveness is more obvious, forming a promotional role with superimposed characteristics, thus stimulating the rise of the core competitiveness of enterprises to a greater extent. (iii) The digital transformation of state-owned steel enterprises and high government subsidy steel enterprises has a more significant effect on improving core

**Table 12. Business efficiency.**

| VARIABLES | (1) CORE | (2) CORE | (3) CORE | (4) CORE |
|---|---|---|---|---|
| did | -0.390** | -0.650*** | -0.318** | -0.701*** |
| | (-2.55) | (-4.17) | (-2.10) | (-4.57) |
| Efficiency | 0.410*** | 0.288*** | 0.073 | -0.112 |
| | (6.66) | (3.03) | (0.71) | (-0.79) |
| did#Efficiency | 0.740*** | 0.799*** | 0.638*** | 0.827*** |
| | (5.31) | (5.60) | (4.53) | (5.88) |
| Constant | -0.449*** | -0.271* | 0.554 | -5.207*** |
| | (-5.26) | (-1.94) | (0.72) | (-3.04) |
| Observations | 547 | 547 | 547 | 547 |
| R-squared | 0.189 | 0.283 | 0.237 | 0.337 |
| Number of Ids | 51 | 51 | 51 | 51 |
| Controls | NO | NO | YES | YES |
| Company FE | NO | YES | NO | YES |
| Year FE | NO | YES | NO | YES |

z-statistics in parentheses

*** $p<0.01$

** $p<0.05$

* $p<0.1$

competitiveness than that of personal enterprises and low subsidy enterprises. (v) In the digital transformation process, enterprises' management capability and operation efficiency, and environmental uncertainty positively improve the results of enterprise digital transformation.

Based on the above research, we provide the following implications.

Enterprises should first pay attention to the role played by management ability and business efficiency in digital transformation. Existing research has found that not all digital transformation models improve the core competitiveness of enterprises. Therefore, enterprises cannot blindly implement digital transformation to avoid falling into the "digital transformation performance trap". Digital transformation is about not only introducing digital equipment but also enterprises' need to cultivate and improve management ability and operational efficiency that are compatible with digital technology. Enterprises can improve their digital technology management capabilities by selecting managers with leadership in the digital arena, cultivating employees with digital thinking, and building digital transformation teams. Enterprises with low operational efficiency should slow the pace of digital transformation and focus first on improving their operational efficiency.

Second, we found that digital transformation has a more obvious promotional effect on the core competitiveness of the next phase, with a cumulative effect of long-term superposition. Therefore, enterprises should not be eager to achieve success but should focus on the long term and patiently carry out digital transformation activities. At the same time, enterprises in backward and leading positions in the industry should actively carry out digital transformation, have the same group effect in the industry, and attract enterprises in the middle of the industry to carry out transformations.

Policy-makers should first strengthen the input of government subsidies to enhance the promotional effect of digital transformation, increase the core competitiveness of enterprises, and strengthen the position of Chinese enterprises in the world market. Second, China should actively comply with the trend of the rapid development of digital technology, fully grasp

opportunities offered by enterprise digital transformation, give strong policy preference to enterprises, encourage the deep integration of digital technology and enterprises in terms of products and organizational structure, and help enterprises develop at a high quality. Enterprises' digital development should follow the principle of differentiation, develop distinctive digital paths according to the special conditions of different enterprises, guide enterprises to adapt their technological innovation and digital transformation needs through "learning by doing", and reduce enterprise risks as much as possible during the integrated innovation process.

However, there are still some deficiencies that also provide useful ideas for future study. First, we only selected the steel industry for the research. In future research, the type of enterprise digital transformation effect that is more significant for all industries can be explored. Second, the measurement of core competitiveness only focuses on enterprises' financial data; however, enterprise core competitiveness is not only reflected in quantitative aspects. Future research can attempt to comprehensively measure enterprise core competitiveness from both quantitative and qualitative aspects. Moreover, due to the limitations of the selected methods, the black box process of how digital transformation affects core competitiveness has not yet been discussed. Future research can consider exploring this process to improve the comprehensiveness of the conclusions.

## Author Contributions

**Conceptualization:** LiuYang Zhang, Peng Cao.

**Data curation:** LiuYang Zhang.

**Funding acquisition:** LiuYang Zhang.

**Methodology:** PingQian Qiu.

**Project administration:** PingQian Qiu, Peng Cao.

**Resources:** PingQian Qiu.

**Supervision:** Peng Cao.

**Writing – original draft:** LiuYang Zhang.

**Writing – review & editing:** Peng Cao.

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
