## [Decision Letter · Decision Letter 0]

12 Jun 2023

PONE-D-23-15731Does digital transformation enhance the core competitiveness?--evidence from Chinese traditional manufacturingPLOS ONE

Dear Dr. Zhang,

Thank you for submitting your manuscript to PLOS ONE. After careful consideration, we feel that it has merit but does not fully meet PLOS ONE’s publication criteria as it currently stands. Therefore, we invite you to submit a revised version of the manuscript that addresses the points raised during the review process.

We look forward to receiving your revised manuscript.

Kind regards,

Syed Abdul Rehman Khan, PhD

Academic Editor

PLOS ONE

Journal Requirements:

2. Please upload a copy of Figure 5, to which you refer in your text on page 13. If the figure is no longer to be included as part of the submission please remove all reference to it within the text.

Reviewers' comments:

Reviewer's Responses to Questions

**Comments to the Author**

1. Is the manuscript technically sound, and do the data support the conclusions?

Reviewer #1: Partly

Reviewer #2: Yes

2. Has the statistical analysis been performed appropriately and rigorously? 

Reviewer #1: No

Reviewer #2: Yes

3. Have the authors made all data underlying the findings in their manuscript fully available?

Reviewer #1: Yes

Reviewer #2: Yes

4. Is the manuscript presented in an intelligible fashion and written in standard English?

Reviewer #1: Yes

Reviewer #2: Yes

5. Review Comments to the Author

Reviewer #1: The reviewer believes that the topic “Does digital transformation enhance the core competitiveness?--evidence from Chinese traditional manufacturing” is worthy of investigation. However, the following needs to be addressed. There are minor and major issues that should be corrected. I believe the paper could be further strengthened by added information about.

Please reorganize the manuscript at the journal request. Please change the reference format.

The language of this manuscript is very bad and needs help from native speakers.

The title of the manuscript should fully demonstrate the content of this study and the relevant subjects.

Abstracts should include the purpose and findings of the study.

PARAGRAPH. Digital transformation is an important measure......... This a very vague statement. These sentences do not provide any information on how the concept could be conceptualized?

PARAGRAPH. Therefore, the discussion......... This section should explain the study's context and research objective. Furthermore, the research gap needs to be narrowed after analyzing the previous studies. The research method is not adequately explained in the first section.

-Introduction, what authors wanted to convey. Here author must build research gap following the previous studies.-The manuscript does not answer the following concerns: Why is it timeliness to explore such a study? What makes this study different from the previously published studies? Are there any similarly findings in line with the previously published studies? Are the findings different from prior academic studies that were conducted elsewhere, if any? For example, information Innovation and Innovation Ecosystems, what it requires, what are the new technologies, some recent issue highlights the importance. See the following: Enhancing Digital Innovation for the Sustainable Transformation of Manufacturing Industry: A Pressure-State-Response System Framework to Perceptions of Digital Green Innovation and Its Performance for Green and Intelligent Manufacturing

. https://doi.org/10.3390/systems10030072

-Methodology: Model.. I suggest authors here build your main heading on Research and data methodology. Clearly explain the model building process, and what previous studies have used similar models (model testing approach).

There is no flow in the text. It partly depends on the lack of proofreading but also on the fact that many statements and claims are made without being followed up by a clear and logical discussion. It is especially problematic in the Introduction that brings up a number of findings from different areas without linking them together.

Please make sure your conclusions' section underscores the scientific value-added of your paper, and/or the applicability of your findings/results. Highlight the novelty of your study.

In addition to summarizing the actions taken and results, please strengthen the explanation of their significance. It is recommended to use quantitative reasoning comparing with appropriate benchmarks, especially those stemming from previous work. See the following: How to Improve the Quality and Speed of Green New Product Development? Processes 2019, 7, 443. https://doi.org/10.3390/pr7070443

More importantly, the choice of the variables should be explained in light of the theory and the prior literature on the topic. The arguments are simply relationships and causes very close to the replication of many studies dealing with the same thing.

The authors should emphasize the important role of digital technology in green innovation in future research. Please consider this structure for manuscript final part.

-Discussion

-Conclusion

-Managerial Implication

-Practical/Social Implications

-Discussion needs to be a coherent and cohesive set of arguments that take us beyond this study in particular, and help us see the relevance of what authors have proposed. Authors should create an independent “Discussion” section. Author need to contextualize the findings in the literature, and need to be explicit about the added value of your study towards that literature. Also other studies should be cited to increase the theoretical background of each of the method used. Findings should be contextualized in the literature and should be explicit about the added value of the study towards the literature (An adoption-implementation framework of digital green knowledge to improve the performance of digital green innovation practices for industry 5.0, https://doi.org/10.1016/j.jclepro.2022.132608.). Limitations and future research.

As any emprical study that use different approaches I would like to ask to introduce in the Conclusion section at least a paragraph containing the study limitations. I noticed some things in the paper but a synthesis of statements related to how the study is useful (or partially useful, since are required certain further analysis) and helps potential interested readers does not really exist. Maybe in addition to the last section of Conclusion it is beneficial to introduce a section called: Discussion.

Reviewer #2: It is a highly fascinating subject. The concept is emerging and might be helpful in today's actual world, specifically in the supply chain and technological transformations field. Therefore, I recommend this article for further process based on the paper's relevance, originality, need, and suitability for the journal. The introduction is skillfully crafted. However, the author may guarantee the order in which the concepts are explained and presented.

The literature review is skillfully prepared and effectively divides various topics with appropriate titles. Every hypothesis development is discussed in the literature, and it is a good idea to describe the theoretical framework after presenting the image. Similar to how the approach is taught, it is presented logically but still needs to be well integrated into the paper.

In addition, I wonder if the authors should revise their literature and incorporate more pertinent literature. A few of the pertinent articles I am providing to the author that must be incorporated into this study are listed below:

• Khan, S.A.R., Ahmad, Z., Sheikh, A.A. and Yu, Z. (2023), "Green technology adoption paving the way toward sustainable performance in circular economy: a case of Pakistani small and medium enterprises", International Journal of Innovation Science, Vol. ahead-of-print No. ahead-of-print. https://doi.org/10.1108/IJIS-10-2022-0199

• Rehman Khan SA, Ahmad Z, Sheikh AA, Yu Z. Digital transformation, smart technologies, and eco-innovation are paving the way toward sustainable supply chain performance. Science Progress. 2022;105(4). doi:10.1177/00368504221145648

• Khan, S. A. R., Tabish, M., & Zhang, Y. (2023). Embracement of industry 4.0 and sustainable supply chain practices under the shadow of practice-based view theory: ensuring environmental sustainability in corporate sector. Journal of Cleaner Production, 398, 136609.

Last but not least, ensure all references are accurate and include all relevant information, such as the volume, issue, and page numbers. Well, the study complied with the requirements set forth by the esteemed publication by providing all pertinent data and by covering all essential topics. To ensure optimum refinement, the author should double-check each line while considering the sample article from a prominent publication.

6. PLOS authors have the option to publish the peer review history of their article (what does this mean?). If published, this will include your full peer review and any attached files.

Reviewer #1: No

Reviewer #2: **Yes: **Dr. Adnan Ahmed Sheikh

---

## [Author Response · Author response to Decision Letter 0]

5 Jul 2023

Dear reviewers

Thank you very much for your comments and suggestions. We have replied to all of your suggestions and attached them in Response to reviewers. We hope that the modifications we have made will meet your suggestions and we are also very willing to make further modifications. Your comments have greatly improved the quality of the article. Thank you again for the time you have spent on our paper, we really appreciate it.

With regards

LiuYang Zhang

---

## [Decision Letter · Decision Letter 1]

17 Jul 2023

Does digital transformation enhance the core competitiveness?--Quasi-natural experimental evidence from Chinese traditional manufacturing

PONE-D-23-15731R1

Dear Dr. Zhang,

We’re pleased to inform you that your manuscript has been judged scientifically suitable for publication and will be formally accepted for publication once it meets all outstanding technical requirements.

Kind regards,

Syed Abdul Rehman Khan, PhD

Academic Editor

PLOS ONE

Reviewers' comments:

Reviewer's Responses to Questions

**Comments to the Author**

1. If the authors have adequately addressed your comments raised in a previous round of review and you feel that this manuscript is now acceptable for publication, you may indicate that here to bypass the “Comments to the Author” section, enter your conflict of interest statement in the “Confidential to Editor” section, and submit your "Accept" recommendation.

Reviewer #1: (No Response)

Reviewer #2: All comments have been addressed

2. Is the manuscript technically sound, and do the data support the conclusions?

Reviewer #1: (No Response)

Reviewer #2: Yes

3. Has the statistical analysis been performed appropriately and rigorously? 

Reviewer #1: (No Response)

Reviewer #2: Yes

4. Have the authors made all data underlying the findings in their manuscript fully available?

Reviewer #1: (No Response)

Reviewer #2: Yes

5. Is the manuscript presented in an intelligible fashion and written in standard English?

Reviewer #1: (No Response)

Reviewer #2: Yes

6. Review Comments to the Author

Reviewer #1: The manuscript has significantly improved as compared to the previous version. Indeed, the authors tried to improve it, and the main weaknesses are solved.

Thus, in my opinion, the manuscript is recommendable for publication.

Reviewer #2: Authors have significantly incorporated the required changes. Before proceeding further, I will suggest the authors to please recheck the grammatical errors and sentence structure

7. PLOS authors have the option to publish the peer review history of their article (what does this mean?). If published, this will include your full peer review and any attached files.

Reviewer #1: No

Reviewer #2: **Yes: **Dr. Adnan Ahmed Sheikh

---

## [Editor Report · Acceptance letter]

20 Jul 2023

PONE-D-23-15731R1 

Does digital transformation enhance the core competitiveness?--Quasi-natural experimental evidence from Chinese traditional manufacturing 

Dear Dr. Zhang:

I'm pleased to inform you that your manuscript has been deemed suitable for publication in PLOS ONE. Congratulations! Your manuscript is now with our production department. 

Kind regards, 

on behalf of

Dr. Syed Abdul Rehman Khan 

Academic Editor

PLOS ONE